# Lipid-Based Molecules on Signaling Pathways in Autism Spectrum Disorder

**DOI:** 10.3390/ijms23179803

**Published:** 2022-08-29

**Authors:** Kunio Yui, George Imataka, Shigemi Yoshihara

**Affiliations:** Department of Pediatrics, Dokkyo Medical University, Mibu 321-0293, Tochigi, Japan

**Keywords:** autism spectrum disorder, lipid modified signaling pathway, endocannabinoid system, prostaglandins, cyclooxygenase-2

## Abstract

The signaling pathways associated with lipid metabolism contribute to the pathophysiology of autism spectrum disorder (ASD) and provide insights for devising new therapeutic strategies. Prostaglandin E2 is a membrane-derived lipid molecule that contributes to developing ASD associated with canonical Wnt signaling. Cyclooxygenase-2 plays a key role in neuroinflammation and is implicated in the pathogenesis of neurodevelopmental diseases, such as ASD. The endocannabinoid system maintains a balance between inflammatory and redox status and synaptic plasticity and is a potential target for ASD pathophysiology. Redox signaling refers to specific and usually reversible oxidation–reduction reactions, some of which are also involved in pathways accounting for the abnormal behavior observed in ASD. Redox signaling and redox status-sensitive transcription factors contribute to the pathophysiology of ASD. Cannabinoids regulate the redox balance by altering the levels and activity of antioxidant molecules via ROS-producing NADPH oxidase (NOX) and ROS-scavenging superoxide dismutase enzymes. These signaling cascades integrate a broad range of neurodevelopmental processes that may be involved in the pathophysiology of ASD. Based on these pathways, we highlight putative targets that may be used for devising novel therapeutic interventions for ASD.

## 1. Introduction

Autism spectrum disorder (ASD) is a set of heterogeneous neurodevelopmental conditions [1] associated with a strong genetic component and involving many risk factors [2]. Genes affect synaptic development, leading to neurobiological theories focusing on the connectivity and neural effects of gene expression [2]. There are few useful non-curative treatments. Improved strategies for early identification with phenotypic characteristics and biological markers might hopefully improve the effectiveness of treatment [3].

The pathophysiology of ASD includes alterations in neuronal circuits induced by defects in various signaling pathways. Most studies on ASD pathogenesis have indicated a combination of genetic mutations and various altered signaling pathways, such as those involving mTOR [4], Wnt, Prostaglandin E2 (PGE2) signaling [5], and endocannabinoid signaling [6]. Although significant progress has been made in understanding the pathogenesis and etiology of ASD, few effective therapies exist. Hence, further research is imperative to identify potential therapeutic targets.

The combination of various core mutations and consequent dysfunction of cell signaling pathways has already been reported in the pathogenesis of ASD in preclinical animal models and patient samples. In this review, we highlight putative therapeutic targets based on these pathways. These putative targets may be used for novel therapeutic interventions for ASD in the future. Timely, specific, and accurate modulation of these pathways may contribute to our understanding of ASD pathogenesis. This review provides new perspectives on the various cellular signaling pathways involved in the regulation of lipid metabolism in ASD pathogenesis.

## 2. Signaling Pathways Associated with Lipid Metabolites

Lipids act as regulatory molecules that contribute to the growth and maintenance of brain functions. Altered fatty acid metabolic pathways may be involved in ASD pathogenesis. A natural lipid-derived molecule, prostaglandin E2 (PGE2), is involved in the physiological processes implicated in ASD [6,7]. Esterified omega-6 polyunsaturated fatty acids (PUFAs), such as arachidonic acid (AA), present on the inner surface of the cell membrane, are converted by phospholipase A2 (PLA2) into the free form, which is further metabolized by cyclooxygenases (COXs) and lipoxygenases (LOXs) into bioactive mediators, such as prostanoids and leukotrienes (LTs) [8]. These mediators have been proposed as novel preventive and therapeutic targets for inflammatory diseases, including some types of ASD [8]. Disrupted COX2/PGE2 signaling has been reported in ASD pathology [9]. PGE2 treatment induces the expression of Wnt target genes [7]. In neuroectodermal (NE-4C) stem cells, PGE2 interacts with canonical Wnt signaling through PKA and PI-3K, which play roles in their migration and proliferation in the brain via the expression levels of Wnt-target genes being upregulated in response to PGE2, inducing crosstalk between PGE2 and Wnt signaling in neuronal cells [7]. Moreover, PGE2 acts via these kinases to converge with the Wnt pathway during prenatal development [7]. These mechanisms could be possible pathogenetic factors underlying neurodevelopmental disorders such as ASD [7].

Additionally, PGE2 receptor EP4 signaling may be part of the neuroprotective and anti-inflammatory pathways in models of neurodegenerative disease [10]. Various genetic and environmental factors influence PGE2 levels and increase the risk of developing ASD [11].

Apart from these, the endocannabinoid system (ECS) may be involved in ASD pathophysiology [12]. The ECS is a complex network of lipid signaling pathways consisting of AA, 2-arachidonoylglycerol-derived compounds, and their G protein-coupled receptors (cannabinoid receptors CB1 and CB2) [13] Moreover, ECS is a lipid cell signaling system involved in the physiology and homeostasis of the brain. Synaptic plasticity, neuroendocrine functions, reproduction, and the immune response also require proper functioning of the ECS [14]. Arachidonic acid-related signaling pathways are presented in Figure 1.

## 3. Prostaglandin E2 (PGE2)

### 3.1. Characteristics

Prostaglandin E2 (PGE2) is a well-known membrane-derived lipid signaling molecule that plays an important role in neuronal development. Abnormal PGE2 levels are due to environmental insults during prenatal development [5,15]. Various genetic and environmental factors can influence PGE2 levels, thereby increasing the risk of developing ASD [11]. Altered PGE2 signaling is associated with the pathological changes observed in the nervous system [7]. Furthermore, PGE2 is associated with canonical Wnt signaling through the protein kinase A (PKA), phosphatidylinositide 3-kinase (PI-3K), and protein kinase C (PKC) pathways [7]. Enhanced levels of PGE2 increase Wnt-dependent motility and the proliferation of neuroectodermal stem cells [8]. PGE2 can influence the expression of genes from the canonical Wnt signaling pathway via β-catenin, which is the major downstream regulator of Wnt-dependent gene transcription [11]. These cellular mechanisms may contribute to ASD development [11].

### 3.2. PGE2 in ASD

Cyclooxygenase-2 (COX-2) is the main regulator of PGE2 synthesis [9]. Abnormal COX-2/PGE2 signaling is associated with behavioral symptoms of ASD [6]. Male mice lacking cyclooxygenase-1 and cyclooxygenase-2 (COX^−1-/-^ and COX^−2-/-^) exhibit altered COX/PGE2 signaling and, hence, are a potential model of ASD [9].

Collectively, PGE2 upregulates the canonical Wnt signaling pathway, which is associated with the development of ASD. The association between PGE2 and Wnt signaling during prenatal development of the nervous system, where PKA and PI-3K might act as mediators, suggests that PGE2 alters Wnt-dependent migration and proliferation of neuroectodermal stem cells, which has implications for ASD [7].

### 3.3. Perspectives on ASD Therapy

Neurodevelopmental disorders, such as ASD, are associated with increased activity of the brain catalase of phospholipase A2 (PLA2) isoforms. As PGE2 biosynthesis has been associated with PLA2 expression [16], new synthetic inhibitors of PLA2 have been used for treating neuroinflammation-associated neurodevelopmental disorders in cells [17].

## 4. Cyclooxygenase-2 (COX2)

Cyclooxygenase-2 (COX-2) plays a key role in neuroinflammation and the pathogenesis of neurodegenerative diseases such as ASD [18]. Endogenous cannabinoids are substrates for COX-2 and can be oxygenated by COX-2 to form new classes of prostaglandins [18]. COX-2 in synaptic signaling may provide a mechanistic basis for designing new drugs aimed at preventing, treating, or alleviating neuroinflammation-associated neurological disorders [18]. For reference, accumulating evidence supports an association between the neuropsychiatric disorders involving ASD and inflammation, and thus anti-inflammatory agents, such as the COX-2 inhibitors (celecoxib), may contribute to a novel avenue for the prevention and treatment of neuropsychiatric illness [19]. The brain morphologic aberrations in ASD may be a result of neural pruning processes and neuroinflammation. Treatment with celecoxib combined with risperidone in a 10-week trial in 40 patients diagnosed with ASD induced efficacy according to the Aberrant Behavior Checklist—Community Edition (ABC-C) in the domains of irritability, social withdrawal, and stereotypy in children with ASD [19]. Autistic children display immune system dysregulation and show an altered immune response of peripheral blood mononuclear cells (PBMCs). The mRNA level for cannabinoid receptor type 2 (CB2) was significantly increased in peripheral blood mononuclear cells in ASD children as compared to healthy subjects. Protein levels of CB-2 were also significantly increased in autistic children. These findings indicate CB2 receptor as potential therapeutic target for the pharmacological management of ASD [20].

COX-2 knockout mice exhibited dysregulated expression of several ASD-related genes [9]. COX-2 deficient mice, lacking the activity of the PGE2-producing enzyme, display ASD-linked behaviors such as hyperactivity, repetitive behaviors, altered motor strength, and awkward social interaction [6].

## 5. The Association between PGE2 and COX-2 in ASD

Regulation of PGE2 synthesis via the modulation of COX-2 activity has been implicated in the pathogenesis of neurodevelopmental diseases such as ASD [9]. Cyclooxygenase-2 (COX-2) is the main regulator of PGE2 synthesis [9]. Abnormal COX-2/PGE2 signaling is associated with behavioral symptoms of ASD [6]. Alteration in the COX-2/PGE2 pathway induces differential expression of ASD-related gene [9].

Irregular COX-2/PGE2 signaling leads to ASD-related behaviors and social abnormalities. Thus, COX-2 may be related to ASD-like behavioral symptoms in association with PGE2. The associations between COX-2 and PGE2 are presented in Figure 2. Of note, prostaglandin families (prostaglandin J2) are endogenous toxic products of cyclooxygenases and are actively involved in the neuronal dysfunction induced by pro-inflammatory stimuli. The production of PGE2 and upregulation in the cyclooxygenase (COX) pathway were induced by PLA2. The arachidonic acid (AA) production and participation of intracellular PLA2 in these events were also evaluated [21].

As PGE2 plays an important role in brain development, abnormalities in the COX-2/PGE2 signaling pathway due to genetic or environmental causes are linked to ASD [22] and may result in autism-related behaviors, such as increased hyperactivity, anxiety, repetitive behavior, motor deficits, and social abnormalities, as observed in several cases of ASD in the developing brain [9].

Cyclooxygenase-2 (COX-2) is the main regulator of PGE2 synthesis [9]. Abnormal COX-2/PGE2 signaling is associated with behavioral symptoms of ASD [9]. Male mice lacking cyclooxygenase-1 and cyclooxygenase-2 (COX^−1-/-^ and COX^−2-/-^) exhibit altered COX/PGE2 signaling and, hence, are a potential model of ASD [9].

With respect to pathophysiology of ASD, abnormal COX-2/PGE2 signaling is associated with behavioral symptoms of ASD [9]. Male mice lacking cyclooxygenase-1 and cyclooxygenase-2 (COX^−1-/-^ and COX^−2-/-^) exhibit altered COX-2/PGE2 signaling and, hence, are a potential model of ASD [9].

Significantly higher levels of both COX-2 and PGE2 have been reported in the plasma samples of autistic patients. Increased plasma PGE2 levels were concomitant with significantly lower levels of α-synuclein [23], which plays a critical role in synaptic functions, including synaptic pool preservation, vesicular stabilization, and synaptic plasticity [23]. The combination of α-Syn, COX-2, and PGE2/its EP2 receptor may be a predictive diagnostic biomarker of ASD in relation to mitochondrial dysfunction and glutamate excitotoxicity as etiological mechanisms of ASD, indicating that the above-described diagnostic markers might contribute to early intervention [23].

In summary, combining α-synuclein, COX-2, and the PGE2/PEA receptor could play an important role in brain development, with abnormalities in the COX-2/PGE2 signaling pathway due to genetic or environmental causes linked to ASD [22].

## 6. Endocannabinoid Signaling

### 6.1. The Role of Endocannabinoids in Brain Function

The endocannabinoid system (ECS) consists of endogenous cannabinoids and metabolic enzymes that play a critical homeostatic role in modulating polyunsaturated omega fatty acid (PUFA) signaling to maintain a balanced inflammatory and redox state [24].

PUFAs may be linked to the ECS and involved in the maintenance of brain plasticity, memory and learning, blood flow, and the genesis of neural cells [25]. The ECS is a pervasive neuromodulatory system that plays an important role in the development of the central nervous system, synaptic plasticity, the response to endogenous and environmental insults [24,26], neurotransmission, and microglial activation [27]. Cannabigerol, one of the major phytocannabinoids present in *Cannabis sativa* L., may regulate signaling and exert therapeutic effects via cannabinoid receptors [27]. ECS’s action is mediated by CB1 and CB2 receptors, which form heteromeric complexes (CB1–CB2 heteroreceptor complexes) [27]. ECS-induced metabolic outcomes may support key mediators and inflammatory resolution pathways critical for maintaining homeostasis [25].

ECS signaling regulates proliferation, differentiation, and the death of brain cells and has important consequences for neural development and brain repair via cannabinoid (CB) receptors [28]. Furthermore, crosstalk of the brain ECS with the immune system maintains the neuroprotective and anti-inflammatory actions [28]. Thus, ECS-mediated maintenance of synaptic plasticity and neuronal cell function may contribute to endogenous stem cell-based neuroprotective strategies [28].

The two cannabinoid receptors, CB1 and CB2, are derivatives of PUFAω-6 AA and ω-3 fatty acids Wu [23]. The ECS plays a role in modulating behavioral traits [29]. The modulation of anandamide (AEA), one of the main endocannabinoids in the brain, has been shown to regulate social behaviors in animal models of ASD [29]. Social deficits, repetitive behaviors, and abnormal emotional behaviors in VPA-treated offspring improved after treatment with an inhibitor of AEA-degrading enzyme [29].

### 6.2. The Role of eCBs in the Development of Brain

As described above, the main endocannabinoid in the brain, AEA, has been reported to alter social behaviors in genetic models of ASD [29]. Such effects were mediated by enhancing the mechanism of removal of postsynaptic α-amino-3-hydroxy-5-methyl-4-isoxazolepropionic acid receptor (AMPAR) subunits GluA1/2 underlying AEA signaling in the PFC in a VPA-induced model of ASD [29]. eCBs exert epigenetic effects on neurotransmitter signaling and regulate their expression. Moreover, eCBs play a role in epigenetic modifications, such as DNA methylation and histone modifications [30]. As eCBs play an important role during neurodevelopment, the interaction between the ECS and epigenetic modifications may play a regulatory role in neurodevelopmental disorders [30]. Thus, eCB can epigenetically modulate neurodevelopment [30]. A recent study based on a murine model of ASD (*Shank3B^–/–^* mice showing substantial social interaction impairment) showed that BLA-NAc glutamatergic circuit activation reduces sociability and social reward seeking without increased anxiety or rewarding effects [31].

2-arachidonoylglycerol (2-AG)-mediated cannabinoid signaling regulates the BLA–NAc glutamatergic circuit [31], which was thus shown to be important for social function in this animal model. 2-AG may be a promising target for the treatment of social domain symptoms in ASD [31].

Studies based on animal models have revealed important roles for ECS combined with G protein-coupled cannabinoid receptors. Furthermore, ECS’s endogenous lipid-derived agonists may contribute to social anxiety and social reward as two key aspects of social behavior [32]. Functional MRI studies in humans support such a role of anandamide [32]. A previous clinical study reported that 18 autistic patients aged between 7 and 18 years were treated with cannabidiol (CBD) and CBD-enriched cannabis sativa extract, including 0.04 to 0.12 mg/kg/day CDB, for 6–9 months. CBD-enriched CE ameliorated multiple ASD symptoms in 30% of patients [33].

Collectively, the role of eCB signaling may be related to social impairment in neuropsychiatric diseases such as ASD. With respect to the therapeutic potential of eCBs, the ECS plays a key role in neurodevelopment via inflammatory responses and, thus, the renowned anti-inflammatory properties of cannabinoids might be useful in treating ASD pathology [32].

### 6.3. The Role of Cannabinoid Compounds of Cannabigerol, CB1 and CB2 Receptors, and Cannabidivarin in ASD Symptoms

Cannabigerol (CBG) is the precursor molecule for the most abundant phytocannabinoids [34]. CBG is one of the major phytocannabinoids in cannabis and acts as a competitive partial agonist ligand [34]. CBG, related to cannabinoid receptors, such as the CB1–CB2 heteroreceptor complex in activated microglia, and CGB are potential targets for the treatment of neurodegenerative diseases [34]. CBG has a potent inhibitory action on oxidative stress, which in turn inhibits IκB-α phosphorylation and the translocation of nuclear factor-κB (NF-κB) [35]. In other words, CBG has a potent inhibitory effect on oxidative stress, acting via downregulation of the expression of the main oxidative markers and preventing IκB-α phosphorylation and translocation of NF-κB via modulation of the MAP kinase pathway [35]. Thus, the antioxidant properties of CBG may help treat oxidative stress-related disorders, such as ASD.

The microglia harbor a completely functional ECS signaling system [12]. In addition, the ECS is closely involved in regulating microglia polarization [12]. Hence, the ECS may affect inflammation due to its role in regulating microglia. The activation of CB2 receptors not only affects the migration, proliferation, and release of proinflammatory cytokines from microglia but also promotes the transformation of microglial cells to the anti-inflammatory phenotype [12].

With respect to the role of CB1 and CB2, CBG binds CB1 and CB2 but functions as a competitive antagonist for CB1. CB1 is important for neuronal differentiation, normal axon migration, and the establishment of neuronal connectivity [12]. Early exposure of lipopolysaccharide (LPS) in rats impairs communication and cognition due to decreased ozendocannabinoid (eCB)–CB1 binding, elevated major endocannabinoid ananadamide (AEA) levels, and increased fatty acid amide hydrolase (FAAH) in the amygdala [12]. Oral administration of the FAAH inhibitor PF-04457845 (1 mg/kg) may normalize LPS-induced social behavior changes [12]. A nonpsychoactive cannabinoid such as cannabidivarin (CBDV) might be beneficial therapeutic effects on social novelty, short-term memory, and hyperactivity in the VPA-induced autism rat model [12]. These neuroprotective effects are mediated by the upregulated expression of CB2 protein in the hippocampus and enhanced activity of microglia. ECS intervention may regulate neuroimmune inflammation via CB2 to alleviate ASD symptoms [12,29]. Exogenous insults, such as exposure to valproic acid, can also dysregulate resting CBD levels and CBD components [29]. Collectively, altered endocannabinoid signaling and immune dysfunction may contribute to ASD pathogenesis and the modulation of social behavior [29].

### 6.4. ASD and Endocannabinoids: The Role of Cannabidivarin and 2-AG

Anandamide-mediated signaling may act selectively in reducing social anxiety and enhancing social rewards in relation to ASD [27]. Importantly, treatment with the cannabinoid agonist WIN 55,212-2 (1.2 mg/kg) persistently hampers social interactions and behavior [29]. According to a recent review article, CBD is considered as a most promising therapeutic target due to its wide range of pharmacological activities involving antioxidant, anti-inflammatory, and neuroprotective properties in neurological and neuroplastic diseases [36]. The endocannabinoid retrogrades signaling pathway is widely expressed in the central nervous system in regulating synaptic plasticity (excitatory and inhibitory). Cannabidiol (CBD)-enriched cannabis extracts have been widely used to treat children with refractory epilepsy [37]. Additionally, CBD was useful as an anxiolytic- and antidepressant-like compound, possibly due to the effects on DNA methylation and microRNAs [38].

Although limited effects of CBD on ASD symptoms, a previous study reported clinical effects of CBD on ASD symptoms [39]. Moreover, according to a former review article, pro-inflammatory cytokines, such as IL-6, IL12, IL-1β, and TNF-α, released by immune cells in the CNS, contribute to the development of neuroinflammation and neurodegeneration [39]. During inflammation, endogenous eCBs, such as arachidonyl ethanolamide (AEA) and 2-arachidonyl glycerol (2-AG), are released from immune cells and neurons in the CNS and play a neuroprotective role. Therefore, eCB is associated with therapeutic effects in the regulation of immune responses. mRNA and CB2 receptors were upregulated in the blood in children with autistic disorder, suggesting the involvement of the ECS in the development of ASD [39].

The phytocannabinoid cannabidivarin (CBDV) has the ability to ameliorate behavioral abnormalities resembling core and associated symptoms of ASD [40]. Restoration of hippocampal eCB signaling and neuroinflammation may contribute to these effects [40]. This ability to restore ECS abnormalities might contribute to its beneficial effects on ASD-like behaviors via the expression of CB2 receptors [38]. As a cellular mechanism, 2-arachidonoylglycerol (2-AG) eCB signaling reduced basolateral amygdala-nucleus accumbens (BLA-NAc), glutamatergic activity, and pharmacological 2-AG augmentation, blocking social interaction deficits [31]. These data reveal the BLA-NAc circuit as a critical regulator of social function and suggest that pharmacological 2-AG augmentation could be a promising approach to the treatment of social domain symptoms in ASDs [31].

Collectively, BLA-NAc circuit activation decreases sociability and is regulated by eCB signaling. NAc-specific 2-AG augmentation and BLA-NAc circuit inhibition normalized social deficits in the Shank3B–/– model of ASDs [31]. Therefore, 2-AG augmentation reduced social deficits in Shank3B–/– mice via normalization of hyperactive GABAergic and glutamatergic signaling in the NAc [31]. These findings reveal the BLA-NAc circuit to be a critical regulator of social function. Importantly, 2-AG augmentation suggests a promising approach to the treatment of social domain symptoms in ASDs [31].

### 6.5. Therapeutic Potential of the Endocannabinoid System for ASD

Cannabis is derived from *Cannabis sativa*, one of the world’s oldest propagated plants, and its extracts are available for medical use [41]. According to a previous review article, cannabis eased hyperactivity, sleep disorders, self-injury, anxiety, behavioral problems, and communication in 188 patients with ASD [41]. A previous clinical trial indicated that a whole-plant cannabis extract containing cannabidiol and Δ9-tetrahydrocannabinol significantly improved total SRS scores in 150 subjects with ASD compared to 36 control subjects who received a placebo (*n* = 36) [42]. According to another study, cannabinoid cannabidivarin (CBDV) alters the balance of striatal ”excitatory–inhibitory” metabolites [41]. ASD symptoms were linked to abnormalities in the striatum and its functional circuitry and, thus, the function of the striatum displayed both hyper- and hypoconnectivity, with numerous cortical regions involved. CBDV shifted atypical connections towards neurotypical findings at the baseline [43]. Therefore, CBDV may play a role in the reduction of hyperconnectivity in ASD.

Interestingly, cannabinoids (CBs) reduced the frequency of seizures in autistic patients via the excitatory/inhibitory imbalance in the excitatory/inhibitory system, such as a CB-related increase in glutamate (the excitatory system) in subcortical regions (i.e., basal ganglia) and a decrease in cortical regions (i.e., dorso-medial prefrontal cortex). CBD increased GABA transmission (the inhibitory system) in critical and subcortical regions of normal development subjects but decreased it in the same areas of the ASD group [44].

Collectively, CB may have important pharmacological potential for the behavioral symptoms associated with ASD and ASD-related seizures. Although they have been shown to reduce seizure frequency via the excitatory/inhibitory system [44], further investigations are warranted to identify the conclusive clinical effects of CB and additional underlying cellular mechanisms.

### 6.6. Endocannabinoid and Valproic Acid-Induced Model of ASD

A previous study reported reduced endocannabinoid (eCB) signaling in children with autism and in an animal model of VAP-induced ASD. 2-AG deficiency induced behavioral defects caused by the absence of its primary synthetic enzyme, diacylglycerol lipase α (DGLα), in dendritic mesoporous silica nanoparticles (dMSNs) [45]. These cellular and circuit-level changes were associated with reduced sociability, excessive repetitive grooming, and social impairment [45]. Furthermore, elevation of 2-AG improved ASD-like features in animal models [46]. Thus, disruption of 2-AG signaling in dMSN circuits could contribute to the social and repetitive behavioral phenotypes associated with ASD [46]. In autistic children and rat models of VPA-induced ASD, increased degradation of enzymes and upregulated expression of cannabinoid receptors may lead to repetitive and stereotypical behaviors, hyperactivity, altered sociability and social preferences, and impaired cognitive functioning [46]. These defects are reportedly improved following acute, as well as chronic, treatment with JZL184 [46], which is the primary enzyme responsible for degrading the eCB2-AG [47]. JZL184 treatment enhanced intrinsic 2-AG levels and ameliorated autistic behavior, such as repetitive and stereotypical behaviors, hyperactivity in the open field test, sociability and social preference in the three-chamber test, and cognitive functioning in VPA-exposed offspring [46]. Thus, JZL184 modulates neurotransmission and neuronal plasticity and may contribute to the pathophysiology of ASD [46]. These findings are important for potential therapeutic targets for ASD. JZL184-induced elevation in 2-AG levels confers protection against neuroinflammation, which ameliorates ASD-like behavior [46]. Thus, reduced CBs were observed in autistic children and a murine model of ASD, boosting 2-AG-ameliorated ASD-like phenotypes, suggesting a novel approach for ASD treatment [46].

## 7. Redox Signaling

### 7.1. The Role of Redox Signaling

Redox signaling refers to the specific and usually reversible oxidation/reduction reactions involved in cellular signaling pathways [48]. Redox signaling regulates several physiological processes (e.g., excitation–contraction coupling) and is involved in a wide variety of pathophysiological, homoeostatic, and stress response pathways [48]. Individuals with ASD exhibit enhanced ROS signaling following phosphatidylinositol 3-kinase/protein kinase 3 (PI3K/AKT) signaling pathway stimulation [49]. Environmentally, cellular levels of reactive oxygen species (ROS) may be affected by prenatal neuroinflammatory dysregulation in neural stem redox signaling [49], suggesting an effect of neuroinflammatory dysregulation on neural stem ROS levels. The genetic mutations were recognized in antioxidant and immune responses and ROS generation [49]. Rescue of maternal inflammatory response revealed that brain overgrowth and pathway activation can occur following inhibition of prenatal nicotinamide adenine dinucleotide phosphate (NADPH) oxidases (NOXs), which are enzymes that generate superoxide or hydrogen peroxide from molecular oxygen [50]. Importantly, environmental factors, such as mild maternal inflammation, and subsequent alterations in the neural stem cell redox balance can significantly change brain development in infants, leading to brain overgrowth and abnormal behaviors associated with ASD [49].

### 7.2. Reactive Nitrogen and Oxygen Species

Reactive oxygen species (ROS) act as secondary messengers regulating redox-sensitive signaling pathways, which elicit specific cellular responses [51]. As redox signaling is an intrinsic component of metabolism in humans, its disruption may contribute to the development of various neuronal disorders [51]. Redox signaling is a tightly regulated process that controls cell growth, differentiation, and death [48]. An imbalance between ROS and oxidative stress may contribute to many pathological conditions, including ASD [49,50]. As S-nitrosylation plays an important role in biological processes in neuronal differentiation and maturation [52], S-nitrosylation-mediated redox signaling may act as a molecular switch, whereby redox-mediated post-translational modifications modulate neurogenesis and neurodegeneration via a common transcriptional signaling cascade [53]. Additionally, ROS act as secondary messengers and affect oxidative activity to influence immune, inflammatory, and other signaling processes and redox-sensitive transcription factors [54]. These mechanisms contribute to reduced antioxidative properties in ASD [54]. The role of ROS in COX-2 and cannabinoids is presented in Figure 3.

### 7.3. The Association between Redox Signaling and Iron

The redox state is predominantly reliant on an iron redox couple and is maintained within strict physiological limits. Iron acts as a redox-active cofactor in many biological processes and plays an important role in cell growth [55]. Iron homeostasis is vital to human health, and iron imbalance can lead to various disorders [56].

A recent study reported that synucleinopathy (the pathologic accumulation of α-synuclein) induces the cellular iron sequestration response, leading to neurobehavioral deficits [55]. As S-nitrosylation plays an important role in biological processes in neuronal differentiation and maturation [52], S-nitrosylation-mediated redox signaling may act as a molecular switch, whereby redox-mediated post-translational modifications modulate neurogenesis and neurodegeneration via a common transcriptional signaling cascade [53].

## 8. Copper Signaling

### 8.1. The Role of Copper Signaling

Human copper transporters are subject of a complex multilayer regulation by various metabolic signals [57]. Ceruloplasmin (Cp) signaling has a broad and growing influence on human health, such as a role in iron homeostasis by protecting tissues from oxidative damage [58]. In mammalian systems, intracellular Cp trafficking is influenced by the redox state that triggers Cp-induced signals [59]. Importantly, copper signaling may prove to be an effective therapy for neurodegeneration [57]. Notably, the Cp-dependent communication between neurons and astrocytes is associated with various brain diseases [60].

### 8.2. Copper Signaling and ASD

The antioxidant capacity of Cp is implicated in various neurodevelopmental disorders, such as ASD. Cp expression in myelinating glial cells is crucial to prevent oxidative stress and neurodegeneration in the central and peripheral nervous systems [61]. Lower plasma Cp levels may contribute to ASD pathophysiology in young individuals [62,63,64]. AA is a known regulator of signal transduction proteins and acts as a precursor of potent signaling molecules [63,64,65]. Importantly, these findings indicate that the signaling mediator Cp levels are downregulated via a competitive association between plasma omega-3 PUFAs and omega-6 PUFAs, such as AA [63,65]. Specifically, lower plasma Cp levels due to lower plasma AA levels may impair various signaling activities.

## 9. Iron Signaling and ASD

A former clinical study indicated that 93 children with ASD had lower ferritin and higher mean corpuscular volume (MCV value) compared to 74 children with no neurodevelopmental disorders other than ASD [66]. The ASD gruop showed significantly higher MCV values due to a deficiency of vitamin B12 and folic acid. Folic acid is essential for the correct synthesis of red blood cells [66]. Vitamin B12 is an essential component for DNA synthesis, and vitamin B12 deficiency causes neurologic and psychiatric symptoms, such as motor dysfunction, sensory and memory deficits, and cognitive impairment. Collectively, ASD patients were more likely to have hypoferritinemia based on deficient vitamin B12 as well as folic acid, inducing ASD-related neurological symptoms [66]. Additionally, a deficiency of iron and vitamin D, as well as iron-deficient anemia, were more common in ASD subjects as compared to control children [67].

## 10. The Association between the Endocannabinoid Systems and Redox Signaling

The evidence indicates an important interplay between the cannabinoid receptors CB1 and CB2, their synthetic and metabolizing enzymes, and various key inflammatory and redox-dependent processes. The ECS may also control oxidative stress by directly or indirectly modulating CB1 and CB2 signaling or CB receptor-independent processes via COX-2-dependent eicosanoid pathways [68]. Modulation of cellular redox homeostasis is induced by the ECS [69].

According to primary findings, cannabinoids (CBs) regulate the redox balance by altering the level and activity of antioxidant molecules [70]. CBs target the regulation of redox-sensitive transcription factors, such as Nrf2 in microglia, indicating an important key role of Nrf2 in initiating the transcription of antioxidant and cytoprotective genes [70]. CB modulates the expression of the antioxidant enzyme heme oxygenase 1, suggesting a role for CBD in regulating cellular ROS levels [70]. CB can also modulate the important antioxidation protein superoxide dismutase and its metabolized activities, such as Cu-, Zn-, and Mn-SOD [70], and upregulate GSH levels, as well as, simultaneously, SOD1 and GPx activity, based on the finding that the administration of CB reduces the oxidized glutathione ratio (GSH/GSSG) in myocardial tissue of diabetic mice and prevents GSH depletion in cardiac tissue following cardiotoxicity [70]. As endocannabinoid (eCB) interrupts free radical chain reactions via the electrophilic aromatic molecular region [70], eCB is closely related to antioxidant capacities. The regulatory impact of eCB on reactive oxygen species (ROS) produces nicotinamide adenine dinucleotide phosphate (NADPH) oxidase and ROS-scavenging superoxide dismutase enzymes [71]. However, few studies have reported this association between cannabidiol (CBD) and SOD in ASD. Future studies should examine these associations in ASD.

Collectively, as described above, changes in neuromodulatory eCB signaling were found in patients with ASD and in VPA-induced rodent models of ASD. The presynaptic release of glutamate induces postsynaptic mGluR-mediated upregulation of eCB and the main endogenous agonists of cannabinoid (CB) receptors, such as anandamide and 2-arachidonoylglycerol (2-AG) [72], indicating a close relationship between the antioxidant glutamine and ECS.

With respect to genetic factors of eCB signaling, a recent study reported that the fatty acid amide hydrolase (FAAH) C385A polymorphism was associated with this signaling in the brain [73]. Epigenetic modifications of the eCB may be related to DNA methylation or histone acetylation/deacetylation [73]. Epigenetic modifications of the eCB may be related to DNA methylation and histone acetylation/deacetylation, and FAAH genes encoding the CB1 receptor and FAAH hydrolyzing enzyme play a relevant role in neurological disease pathogenesis and progression [73]. Moreover, FAAH genes may be involved in ASD via network-level changes in neural connectivity associated with genetic variations in endocannabinoid signaling and genotype-associated neural differences [74]. The ECS and ROS are key cellular signaling systems in the modulation of diverse cellular functions. A former review article suggested that CB1 activation and upregulation of brain CB2 receptors reduce oxidative stress in the brain, inducing less tissue damage and less neuroinflammation. Upregulation of CB2 in the peripheral and central nervous systems may reduce neuroinflammation [75]. Furthermore, increased expression of CB1 receptors leads to increased oxidative stress, lipid modifications, and inflammation, which, in turn, may promote the progression of psoriasis into the advanced, arthritic form of the disease [76].

## 11. Modulation of Endocannabinoid Action in Neuroinflammation Is Related to ASD

Neuroinflammation contributes to the pathogenesis of various neurodevelopmental disorders, such as ASD. Microglia and astrocytes are the main pathological components in this process [76]. The endocannabinoid 2-arachidonoylglycerol (2-AG) modulates synaptic function, neurophysiology, and behavior. 2-AG signaling is terminated by monoacyl-glycerol lipase (MAGL) [77]. MAGL is broadly expressed throughout the nervous system, and different brain cell types contribute to the regulation of 2-AG activity. Astrocytes regulate 2-AG content and endocannabinoid-dependent forms of synaptic plasticity and behavior [77]. Astrocytic MAGL is responsible for modulating 2-AG in neuroinflammation through prostaglandins. The astrocytic–neuronal interplay thus provides distributed oversight of 2-AG metabolism and function and protects the nervous system from excessive CB1 receptor activation in neuroinflammatory activation related to ASD [77]. Microglia modulates brain homeostasis by controlling neuronal proliferation/differentiation and synaptic activity [78]. As chronic inflammatory activation of microglia is correlated with several neurodegenerative diseases, the endocannabinoid (eCB) system cannabinoid 2 receptor (CB2R) signaling shifts the balance of neuroprotective genes, and homeostatic genes may acquire therapeutic functionality in microglia for neuroinflammation involved in ASD [79]. Glutamine synthetase (GS), an astrocyte-specific enzyme, plays an important role in neuroprotection and can be modulated by eCB 2-arachidonoylglycerol (2-AG) through extracellular signal-regulated protein kinase 1/2 (ERK1/2) [79]. Microglia modulate brain homeostasis by controlling neuronal proliferation/differentiation and synaptic activity [80]. Chronic inflammatory activation of microglia is correlated with several neurodegenerative diseases, and functional modulation of microglial phenotypes has been considered as a potential therapeutic strategy [78]. The eCB system’s cannabinoid 2 receptor (CB2R) signaling shifts the balance of expression between neuroinflammatory genes, neuroprotective genes, and homeostatic genes toward the latter two, by which microglia acquire therapeutic functionality in the neuroinflammation involved in ASD [79]. The modulation of endocannabinoid action in neuroinflammation in astrocytes and microglia is presented in Figure 4.

## 12. The Association between Lipid Metabolism and the Endocannabinoid System and Its Related Signaling Pathways

The endocannabinoid system (ECS) has been representatively examined in the fields of neuroscience and neurodevelopment. With respect to the pathophysiology of ASD, several studies indicate that eicosanoids are lipid-based signaling molecules that play a role in the pathophysiology of ASD due to their interaction with cyclooxygenase (COX)-derived prostaglandins (PGs) and PGE2 [80]. Although ASD is a complex neurodevelopmental disorder with a multifactorial etiology of signaling pathways, the hypothesis of a brain link between the core behavioral symptoms of ASD and a chronic neuroinflammatory state of the brain may represent the main pathogenesis of ASD. The ECS plays a key role in neurodevelopment and normal inflammatory responses, and may provide new biomarkers and promising therapies in the future [80].

## 13. Dysregulation of the ECS in Fragile X Syndrome

Fragile X syndrome (FXS) is the most common heritable form of mental retardation and monogenic cause of ASD. FXS is characterized by motor, cognitive, and social alterations, mostly overlapping with ASD behavioral phenotypes. The severity of these symptoms and their timing may be exacerbated or advanced by environmental adversity interacting with the genetic mutation [81]. Symptoms of ASD are frequently observed in patients with FXS; however, behavioral similarities and differences between FXS and ASD are important in the causes and correlations of ASD with FXS. Individuals with FXS and comorbid ASD have more severe behavioral problems than individuals with FXS alone; however, patients with FXS and comorbid ASD showed less severe impairments in the social and communication symptoms than patients with nonsyndromic ASD. Subjects with FXS present with anxiety and seizures in addition to comorbid ASD symptoms, and differences in these conditions are recognized in patients with FXS and ASD, possibly due to the difference in the role of fragile X mental retardation 1 protein (FMRP) in FXS and phenotypes of FMRO in ASD [82]. Social anxiety is a common disorder with negative impacts in multiple domains of function in clinical groups including those with fragile X syndrome and ASD. Initial social avoidance characterized all males with FXS, whereas prolonged social avoidance was associated only with boys with FXS who had elevated ASD features [83]. FXS is a leading inherited cause of ASD and intellectual disability, resulting from a mutation in the *FMR1* gene and loss of its protein product FMRP. Although it has this simple genetic origin, FXS is a phenotypically complex disorder including a range of physical and neurocognitive disruptions. Circuit hyperexcitability may be a common convergence point accountable for many wide-ranging phenotypes in FXS [84]. The mechanisms for hyperexcitability in FXS include changes in excitatory synaptic function and connectivity, reducing inhibitory neuron activity and resulting in ion channel expression and conductance. FMRP is also an important regulator of activity-dependent plasticity in the brain and can be a cause and consequence of hyperexcitable networks in FXS [84].

The new drug ZYN002 is a transdermal cannabidiol gel formulation and targets the dysregulation of the endocannabinoid system [85]. Treatment with ZYN002 was found to be safe and well-tolerated in a phase 2 open-label study and extension conducted in 20 children and adolescents with FXS. Behavioral symptoms were improved by blockade of presynaptic CB1 in the mouse model via blockade of the degradation of 2-AG. A blockade of the degradation of anandamide improved learning and memory in the mouse model [85]. Fragile X syndrome (FXS) is caused by the silencing of the *FMR1* gene, leading to the loss of fragile X mental retardation protein (FMRP), a synaptically expressed RNA-binding protein regulating translation. The ECS is a key modulator of synaptic plasticity, cognitive performance, and seizure susceptibility [86]. The cannabinoid receptors CB1 and CB2 are activated by mGluR5 activation-related phospholipid-derived endocannabinoids. Thus, blockage of ECS is a potential therapeutic approach to normalize specific alterations in FXS [86]. As fatty acid amide hydrolase (FAAH) catalyzes the metabolism of the eCB anandamide, treatment with the FAAH inhibitor URB-597 improved performance in fragile X syndrome gene fmr1 KO mice, possibly via the anxiolytic effects of FAAH inhibition. The results indicate that the eCB system is involved in FXS and suggest that the eCB system is a promising target for the treatment of FXS [87]. Collectively, dysregulation of ECS may be a key element in the pathogenesis of FXS.

## 14. Limitation of This Review

This review has several limitations: (1) the discussion of neuroinflammation in general is rudimentary; (2) the clinical use of non-steroid anti-inflammatory drugs and cannabinoids does not support the conclusion.

(1) With respect to the role of neuroinflammation in ASD, neuroinflammation is emerging as a key underlying mechanism of ASD [88]. The various inflammatory and anti-inflammatory effects of dietary components and changes in inflammatory status can prime offspring brain development in neurodevelopmental disorders. Although the impact of maternal nutrients contributes to neuroinflammation changes in offspring, microglia are key immunocompetent cells that produce and respond to inflammatory cues. During neurodevelopment, these cues lead to changes in the neurovascular development and maturation [88]. The regulation of non-neuronal cells and the control of neuroinflammation depend on the local synthesis of the endogenous lipid amide palmitoylethanolamide (PEA) and related endocannabinoids. When the imbalance between the synthesis and degradation of PEA is disrupted, the behavior of non-neuronal cells may not be appropriately regulated, and neuroinflammation exceeds the physiological boundaries [89]. Once administered in formulations with adequate bioavailability, PEA may provide safe and effective control of neuroinflammation [89]. Although the role of inflammation in the underlying mechanisms in specific subgroups/subtypes of ASD (e.g., those with concurrent immunological disorders) has been reported, the role of anti-inflammatory medications as adjuvant therapy is unclear [90]. The exact mechanisms of action, including the biological association between the anti-inflammatory mechanisms and behavior changes of individuals with ASD, are not clear [90]. Robust, large-scale clinical trials are needed to support the efficacy and safety of anti-inflammatory interventions in ASD [90].

(2) With respect to the clinical use of non-steroid anti-inflammatory drugs and cannabinoids, the current evidence is suggested to be insufficient to support the prescription of cannabinoids for the treatment of psychiatric disorders involving ASD [91]

Cannabis has been used to alleviate symptoms associated with ASD (i.e., hyperactivity, irritability, and aggressiveness) [92]. Moreover, cannabis and cannabinoids improved cognition, sensory sensitivity, attention, social interaction, and language [92]. The most common adverse effects were sleep disorders and restlessness [92]. However, comparing the studies’ results is not possible because the authors used different designs to measure the evolution of symptoms through cannabis use in autism, including a systematic review using questionnaires or scales developed by the authors themselves [92]. None of the studies reported the use of neuropsychological assessments to investigate the cognitive aspect [92]. Moreover, samples were small, and several participants were lost during the study period [92]. Additionally, clinical studies should consider recruiting subjects within narrower ranges of age and functional level, assess the long-term tolerability and safety of cannabinoid treatments, and identify target populations within the ASD population that might benefit from these treatments [42].

Additinaly, a recent clinical study conducted in 53 children (median age: 11 years) indicated that cannabidiol may improve symptoms of comorbidities of ASD; however, the long-term effects should be evaluated in a large-scale study [93].

With respect to the risks of endocannabinoid therapy, cannabinoid signaling may be involved in the social impairment and repetitive behaviors observed in individuals with ASD due to a possible role of the EC system in excitatory–inhibitory (E-I) imbalance and immune dysregulation in ASD [94]. Moreover, the consumption of cannabis induces a higher risk of gingival and periodontal disease, oral infection, and cancer of the oral cavity [95]. Therefore, the medical use of endocannabinoids is limited. In summary, the clinical effects of endocannabinoid therapy do not support the conclusion.

## 15. Conclusions

The signaling pathways associated with lipid metabolism help to identify abnormalities and new strategies for treating the pathophysiology of ASD. The available experimental evidence links to several common signaling pathways, such as prostaglandin-E2 (PGE2) and cyclooxygenase-2 (COX-2), endocannabinoids (eCBs), redox signaling, iron signaling, and copper signaling.

The alteration of PGE2 function by environmental insults has been linked to brain pathology, including ASD. PGE2 is a membrane-derived lipid signaling molecule and plays an important role in neuronal development, contributing to the risk of developing ASD associated with canonical Wnt signaling. COX-2 is an enzyme for converting arachidonic acid to prostaglandins and plays a key role in neuroinflammation, being implicated in the pathogenesis of neurodegenerative diseases, including ASD. The endocannabinoid system (ECS) consists of signaling for the maintenance of a balance between inflammatory and redox states, synaptic plasticity, and neuroendocrine functions. The ECS plays a role in epigenetic modifications, such as DNA methylation and neurodevelopment, via inflammatory responses, contributing to the pathophysiology of ASD. Abnormal COX-2/PGE2 signaling is related to the development of ASD. New synthetic inhibitors of lipolytic enzyme phospholipase A2 (PLA2), which is involved in the inflammatory atherosclerotic process, may be used for the treatment of neuroinflammation-associated neurodevelopmental disorders in cells. As described in the limitation section, the clinical use of non-steroidal anti-inflammatory drugs and cannabinoids are limited.

Redox imbalance induces cellular changes, leading to abnormal behaviors in ASD. Redox signaling refers to the specific and usually reversible oxidation/reduction modification of molecules involved in cellular signaling pathways. Environmental factors related to mild maternal inflammation and subsequent alterations in brain development may induce cellular changes, leading to the abnormal behaviors in ASD. The final common pathway of enhanced reactive oxygen species (ROS) signaling may act via resultant PI3K/AKT pathway stimulation. The ECS may also control oxidative stress by directly or indirectly modulating CB1 and CB2 signaling or CB receptor-independent processes via COX-2-dependent eicosanoid pathways. The role of ROS in signaling pathways and redox-sensitive transcription factors contributes to the pathophysiology of ASD.

Cannabinoids regulate the redox balance by altering the level and activity of antioxidant molecules via ROS-producing NADPH oxidase (Nox) and ROS-scavenging superoxide dismutase enzymes.

Neuroinflammation contributes to the pathogenesis of various neurodegenerative diseases, such as ASD. Microglia and astrocytes are main pathological components in this process in relation to the endocannabinoid systems.

We reviewed the current knowledge on the main signaling pathways that are commonly disturbed in ASD. Additionally, we highlighted some putative druggable targets along these pathways in relation to novel therapeutic interventions for ASD. Fragile X syndrome (FXS) is the most common heritable form of mental retardation and monogenic cause of ASD. FXS is a leading inherited cause of ASD and intellectual disability, resulting from a mutation in the *FMR1* gene and loss of its protein product FMRP.

Accurate modulation of these prominent pathways may contribute to the neurodevelopmental regulation of homeostatic patterns and could be used to treat some ASDs.

## Figures and Tables

**Figure 1 ijms-23-09803-f001:**
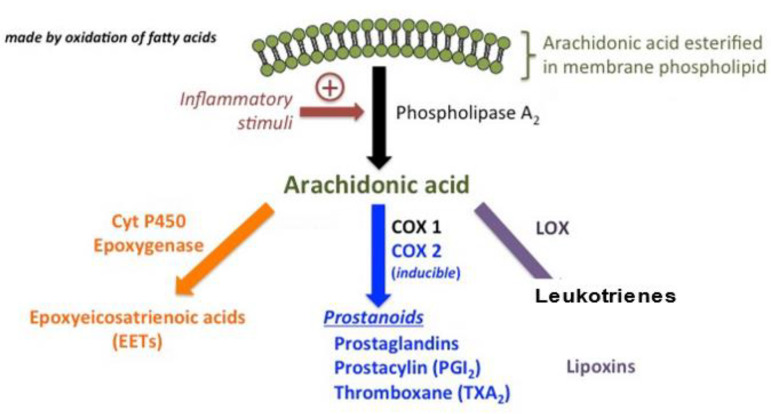
Arachidonic acid is the ordinary substrate for eicosanoid synthesis. The cyclooxygenase (COX) pathways form prostaglandins (PGs) and the COX pathways form leukotrienes (LTs) and the cytP450 pathways form various epoxy, hydroxy and dihydroxy derivatives.

**Figure 2 ijms-23-09803-f002:**
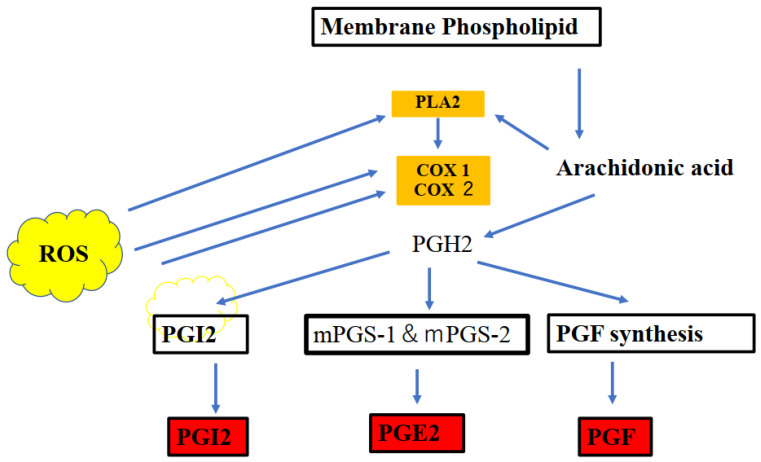
Cyclooxygenases 1 and 2 (COX-1 and COX-2), which are key enzymes in the conversion of arachidonic acid into bioactive prostanoids play a central role in the inflammatory cascade. Prostaglandin families (postaglandin J2) are endogenous toxic products of cyclooxygenases, and because their levels are significantly increased upon brain injury, they are actively involved in neuronal dysfunction induced by pro-inflammatory stimuli. The line of evidence indicates an important interplay between the cannabinoid receptor type 1 (CB1) and cannabinoid receptor type 2 (CB2) and endocannabinoids (ECB) are key inflammatory and redox-dependent processes. The endocannobinoid system (ECS) may also control oxidative stress by directly or indirectly modulating CB1 and CB2 signaling or CB receptor-independent processes via COX-2-dependent eicosanoid pathway. Modulation of cellular redox homeostasis is induced by the ECS. The reactive oxygen species (ROS) act as second messengers and exert oxidative activity to influence immune, inflammatory, and other signaling processes.

**Figure 3 ijms-23-09803-f003:**
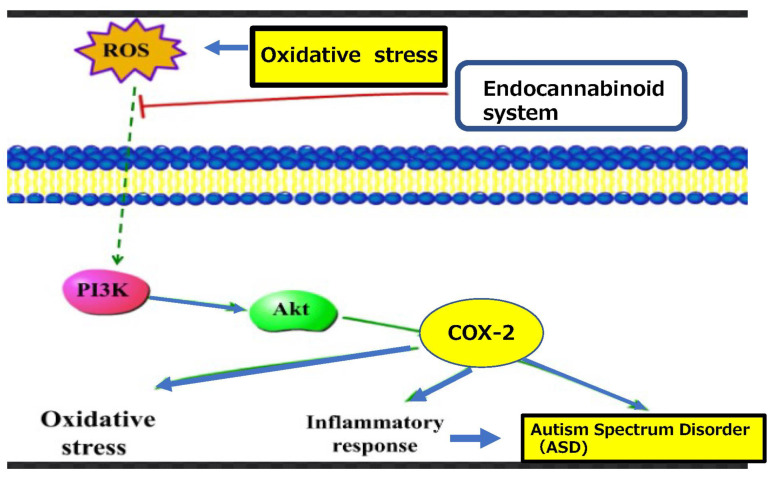
Oxidative stress related oxygen species activate redox signaling which regulates several physiological processes in a wide variety of endocannabinoid syatem in pathophysiological and homoeostatic pathways such as COX-2, inducing neuriinflammation in relation to ASD.

**Figure 4 ijms-23-09803-f004:**
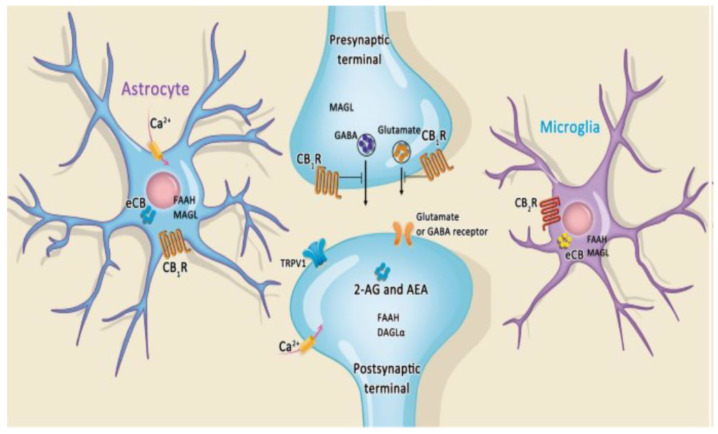
The association among endocannobinoid systems, cannobiboid receptor and GABAnergic intrrneuron. The endocannabinoid system is emerging as a regulator of microglia, resulting in the neuronal-microglia communication system. Cannabinoid 1 (CB1) receptor signaling on Gamma-Amino Butyric Acid (GABA) ergic interneurons plays a crucial role in regulating microglial activity in relation to pro-inflammatory states and neurodegeneration.

## Data Availability

Not applicable.

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
