# Peer review of "Lipid-Based Molecules on Signaling Pathways in Autism Spectrum Disorder"

_ijms, 2022, doi:10.3390/ijms23179803_

Round 1
Reviewer 1 Report
Minor revision
1) Lines 640-641 FXS was caused by a mutation of the fragile X mental retardation 1 (FMR1) gene, resulting in a deficit of fragile X mental retardation protein (FMRP) [90]
Should be updated using a new ACMG nomenclature:
the FMR1 gene “Fragile X messenger ribonucleoprotein 1” gene (the FMR1 should be in italic) and the protein produced by this gene to "Fragile X Messenger Ribonucleoprotein", or FMRP.
2) Lines 642-644
“The clinical presentation of FXS is variable, and is typically associated with developmental delays, intellectual disability, a wide range of behavioral issues, and certain identifying physical features [90].”
This manuscript is focusing on ASD, so it makes sense and the authors are encouraged to mention ASD in FXS and cite it accordingly.
the insertion may look like..
…. a wide range of behavioral issues, including ASD and anxiety in FXS, and certain identifying physical features [90] (additional relevant references..)..
For example, there are recent clinical genotype-phenotype studies that examined ASD, and anxiety, in FXS, and also measured FMRP that was correlated with ASD in FXS.
The relevance is that FMRP tightly regulates the endocannabinoid system, which is downregulated in FXS due to the deficit of the FMRP.
Lines 644-645, the authors do state the role of FMRP citing a preclinical work. “Preclinical evidence reported that fragile X mental retardation protein (FMRP) deficiency enhances mGluR1-dependent endocannabinoid mobilization and subsequent synaptic effects [90].”
Author Response
please see file attached
Reviewer 2 Report
Interesting review article, however some issues need to be further clarified.
- What about therapeutical application for COX as target? Authors cited but not discussed ref.#13.
- While is clear the association between redox and iron, it is not clear the role of iron in ASD.
- This important paper could be added and discussed: DOI 10.1007/s10803-013-1824-9
- Paragraph 11: too long and it seems not related to ASD.
- The paragraph on Fragile X syndrome is not related with ASD and could be removed.
- Figure 3 has low resolution and some errors, i.e. the white box covering "enzyme" word.
- Edit the format, i.e. references in the text should be only by numbers, not names. Typesettings and font also require editing.
Author Response
please see file attached

This manuscript is a resubmission of an earlier submission. The following is a list of the peer review reports and author responses from that submission.
Round 1
Reviewer 1 Report
While this review discusses the impact of several lipid-based molecules on signaling pathways in ASD, this would not be considered “lipid metabolism.” In fact, the authors primarily focus on the role of these lipid-based molecules in neuroinflammation. The title and concept should be re-worked to reflect that. Additionally, there are significant grammatical errors (and some scientific ones) that are probably related to poor English translation; some but not all of these errors are discussed in the following comments. These will definitely need to be fixed prior to publication.
Introduction
- Authors indicate that ASD is a neurodegenerative disease, which is not at all true. They then later discuss potential neurodevelopmental processes underlying ASD pathophysiology. It’s possible that this confusion is due to poor English translation, though this will need to be revised.
- The introduction could benefit from a more thorough description of the clinical findings and societal burden of ASD, as well as what is generally understood about its pathophysiology.
Signaling Pathways Associated with Lipid Metabolites
- ASD is described as an inflammatory disorder, which is highly debatable. While immune pathways likely play a role in some cases of ASD pathogenesis, ASD is not generally considered an inflammatory disorder.
- This section uses the phrase “plays a role” several times without discussing any specific mechanisms and sometimes comes across as a laundry-list of molecules rather than a description of a cohesive system.
- There is an error in the Figure 1 legend where “COX” should be replaced with “LOX.”
PGE2
- Suffers from poor English translation in numerous sentences (first sentence has incorrect verb tense, second sentence is incomplete, etc.).
- There is discussion of the role of COX2 in this section that should probably be moved if these sections are to all remain separate.
COX-2
- Authors again refer to ASD as neurodegenerative.
- The “Genetic Findings” section is confusing.
- Figure 2’s legend refers to “prostaglandin J2,” but only “PGH2” and “PGI2” are represented on the figure; none of these terms were discussed in the text. This figure legend also includes a lot of information about the endocannabinoid system that is not present in the figure itself.
Association Between PGE2 and COX-2
- Information about a-synuclein came out of nowhere.
- This section should be combined with the previous two sections into one cohesive and comprehensive section. Otherwise, it is unclear why COX-2 is discussed in the PGE2 section, and why PGE2 is discussed in the COX-2 section; the sections don’t really seem distinct.
Endocannabinoid Signaling
- There’s an error in the last paragraph of section 6.1: the CB1 and CB2 receptors are not “defined as endocannabinoids.”
- The section on valproic acid is confusing and doesn’t seem to flow from previous information. There is inconsistent use of italics when referring to cannabis strains and mouse strains (like shank3-/-).
- In section 6.5, cannabidiol is misspelled and incorrectly abbreviated as CBG (which was previously defined as cannabigerol) even though the acronym CBD is correctly used later.
- First sentence of second paragraph in section 6.5 has inappropriate capitalization (“Moreover, According…”).
- In several cases, the authors cite “a former review article” when they could be discussing the actual data that they are describing. There is poor English in the final sentences of section 6.6.
- Some abbreviations (like eCB) are inconsistently used and sometimes redefined several times.
Redox Signaling
- This section is a great example of how the review generally focusses on inflammatory processes that happen to use a lipid signaling component rather than focusses on lipid metabolic pathways.
- Figure 3 has strange white boxes on it.
Copper Signaling
- ASD is again referred to as neurodegenerative.
- Section 7.2 has significant English errors that make the science confusing in parts.
- Section 7.2 discusses therapies targeting iron signaling, which should be moved to the next section.
- This section should just be eliminated, and important information can be incorporated into the Redox section.
Iron Signaling
- Significant punctuation errors.
- This section seems completely unrelated to the rest of the review.
Association between Endocannabinoid Systems and Redox Signaling
- CB1 and CB2 are again misidentified as endocannabinoids rather than receptors.
- Authors again reference a review article instead of primary findings.
- Citation structure error in the second paragraph.
- Authors only state in the final sentence of this section that this information has a “relevant role in neurological disease pathogenesis and progress, [including] ASD.” (Including is misspelled as invilving?) If this is the case, they should discuss the specific impact on ASD that each of these mechanisms has.
Modulation of Endocannabinoid Action in Neuroinflammation in related to ASD
- Grammatical error in title of the section
- ASD described as a neurodegenerative disease
- Significant English errors make some sections very difficult to understand
- This section actually discusses significantly less neuroinflammation than the rest of the paper despite its title.
Conclusion
- This review is disorganized and suffers from poor flow.
- The biological processes discussed might be interrelated as are all processes at some level. However, there are major scientific flaws in the content of the review, including an apparent lack of understanding of ASD.
Reviewer 2 Report
A combination of various core mutations and consequent dysfunction of cell signaling pathways have already been reported in the pathogenesis of ASD in preclinical animal models and patient samples. In this review, the authors highlight putative therapeutic targets based on these pathways, focused on the endocannabinoidsystem (ECS), which may be used for novel therapeutic interventions for ASD in the future.
Overall, this is a well written review informs potential readers about the ECS that maintains a balance between the inflammatory and redox status and synaptic plasticity and is a potential target for circumventing ASD pathophysiology.
Minor suggestions:
• Suggest expanding this review to add on a dysregulation of the ECS in Fragile X Syndrome (FXS).
• The functional consequences of reduced or absent the Fragile X Messenger Ribonucleoprotein 1 (FMRP) in people with FXS likely reflect changes in both developmental, including intellectual disability and autism spectrum disorder, and dynamic regulation of multiple intracellular processes 10.1093/med/9780199937837.003.0052, 10.3390/cells11081325
o Absence of FMRP dysregulates the ECS via DAGL, thus disrupting normal ECS function
10.1038/ncomms2045
– FMRP has a recognition sequence for DAGL mRNA10.1038/ncomms2045
– Improper DAGL mRNA trafficking leads to ectopic 2-AG release and the dysregulation of retrograde EC signaling in response to neuronal activity 10.1038/ncomms2045
o Downstream dysregulation of endocannabinoid signaling in the CNS is one proposed mechanism for the clinical abnormalities seen in FXS10.1038/ncomms2045 ,10.1038/nm.3127 and is a promising target for treatment of FXS 10.1016/j.bbr.2015.05.003
o Reductions of FMRP are thought to impair endocannabinoid-mediated regulation of glutamate signaling in FXS 10.1038/ncomms2045
• Clinical Study Of caNNabidiol in childrEn and adolesCenTs With Fragile X were either completed (NCT03614663), whose results are consistent with the proposed mechanisms of action of CBD in FXS* or underway (NCT04977986).
* Elizabeth Berry-Kravis et al., 2022, A Randomized, Controlled, Trial of ZYN002 Cannabidiol Transdermal Gel in Children and Adolescents with Fragile X Syndrome JNDD-D-22-00025 (accepted)
Round 2
Reviewer 1 Report
The authors continue to describe ASD as a neurodegenerative disorder in the abstract.
Introduction has two completely redundant paragraphs.
3.2 PGE2 in ASD
Weird spacing issues (COX-1-/- and COX-2-/-). The knockout symbols should also be superscripted.
4. COX-2
Incorrect citation format in first paragraph.
5. Association between PGE2 and COX2 in ASD
The authors again refer to ASD as neurodegenerative.
First, third, and fourth paragraphs are redundant with section 3.2.
Concluding that COX-deficient mice could be a model for ASD is a bit of a stretch.
6.1 Endocannabinoids in Brain Function
Incorrect citation format in first and third paragraphs.
CB1 and CB2 receptors are written using different nomenclature throughout the section (CB1 vs CB1 vs. CB(1), etc.)
6.2 Genetic Findings
The title of this section is confusing.
Incorrect citation format in third paragraph.
6.3.. The role of cannabinoid compound s of cannabigerol , CB1 and CB2 receptors and cannabidivarin in ASD symptoms
Two periods in section title.
Improper bolding in first paragraph.
6.4.. ASD and Endocannabinoids: the role of cannabidivarin and 2AG
Two periods in section title.
Cites other review articles instead of primary data.
10.Modulationof Endocannabinoid Action in Neuroinflammation is related to ASD
Again, describes ASD as neurodegenerative.
Sections on iron and copper have nothing to do with lipid-based molecules.
Conclusion very quickly recaps main points of the article without adding any sort of discussion, synthesis, or new insights. It actually highlights how disconnected a lot of the sections are.
General:
The article focusses on anti-inflammatory pathways (only some of which involve lipid-based molecules), so why does the title choose to emphasize lipid-based molecules? The authors don’t discuss anything about lipid metabolism or why it matters that most of the molecules they’re talking about happen to be lipid-based.
Abbreviations are inconsistently used and occasionally redefined.
Revised figures and legends are not present and cannot be commented on.
The article still lacks focus and still requires re-organization and a clearer message. While it does review the literature, it doesn’t do so in any novel way that hasn’t already been reviewed, and it doesn’t really contribute any clarity to the field.
Author Response
The authors continue to describe ASD as a neurodegenerative disorder in the abstract.
My answer: I have changed to neurodevelopmental disorder in the Abstract as marker yellow colour.
Introduction has two completely redundant paragraphs.
My answer: I have changed the sentences in the Introduction to concise and non- redundant phrases as marked yellow collar.
3.2 PGE2 in ASD
Weird spacing issues (COX-1-/- and COX-2-/-). The knockout symbols should also be superscripted.
My answer: (COX-1-/- and COX-2-/-) were superscripted as follow (COX-1-/- and COX-2-/-)
- COX-2
Incorrect citation format in first paragraph.
My answer: I corrected citation of reference 19 as follow:
“Cyclooxygenase-2 (COX-2) play a key player in neuroinflammation, and the pathogenesis of neurodegenerative diseases such ASD. [19] Endogenous cannabinoids are substrates for COX-2 and can be oxygenated by COX-2 to form new classes of prostaglandins [19]. COX-2 in synaptic signaling may provide a mechanistic basis for designing new drugs aimed at preventing, treating or alleviating neuroinflammation-associated neurological disorders.[19]”as marked by yellow colours on page 8, Section 4. COX-2.
- Association between PGE2 and COX2 in ASD
The authors again refer to ASD as neurodegenerative.
My answer: I have changed to neurodevelopmental disorder
First, third, and fourth paragraphs are redundant with section 3.2.
My answer: I have deleted sentences similar to section 3.2.
Concluding that COX-deficient mice could be a model for ASD is a bit of a stretch.
My answer: I have deleted the sentences on “COX-deficient mice could be a model for ASD” and added the sentences such as“COX-2 is the main regulator of PGE2 synthesis [9]” and “Alteration in the COX-2/PGE2 pathway induce differential expression of ASD-related gene [9] as marked by yellow colour.
At your suggestion, I have changed the sentence on “Male mice lacking cyclooxygenase-1 and cyclooxygenase-2 (COX-1-/- and COX-2-/-) exhibit altered COX/PGE2 signaling, and hence, are a potential model of ASD [9].” to “Alteration in the COX2/PGE2 pathway induce differential expression of ASD-related gene [9]”marked by yellow colour. In addition, summarized sentences, “being a valuable biomarker of the pathophysiology of ASD [22].” was deleted.
6.1 Endocannabinoids in Brain Function
Incorrect citation format in first and third paragraphs.
My answer: I have changes these sentencers to “The endocannabinoid system (ECS) consists of endogenous cannabinoids, and metabolic enzymes that play a critical homeostatic role in modulating polyunsaturated omega fatty acid (PUFA) signaling to maintain a balanced inflammatory and redox state.
[23].”
CB1 and CB2 receptors are written using different nomenclature throughout the section (CB1 vs CB1 vs. CB(1), etc.)
My answer: I have used CB1 and CB2 throughout the text ( P10-P12) because these abbreviation were frequently used in PubMed.
6.2 Genetic Findings
The title of this section is confusing.
My answer: I have changed the section to “The role of eCB in the development of brain”
Incorrect citation format in third paragraph.
My answer: I have corrected to the paragraph as follow by marked yellow colour; “Such effects are mediated by enhancing the mechanism of removal of postsynaptic α-amino-3-hydroxy-5-methyl-4-isoxazolepropionic acid receptors (AMPARs) subunits GluA1/2 underlying AEA signaling in the PFC in a VPA-induced model of ASD [28].”
6.3. The role of cannabinoid compound s of cannabigerol , CB1 and CB2 receptors and cannabidivarin in ASD symptoms
Two periods in section title.
My answer: I have corrected to one period.
Improper bolding in first paragraph.
My answer: I have corrected improper bolding.
6.4. ASD and Endocannabinoids: the role of cannabidivarin and 2AG
Two periods in section title
My answer: I have corrected to one period
Cites other review articles instead of primary data.
My answer: I have deleted the former sentence “According to a recent review article, CBD may have therapeutic potential in treating neurologic disorders, however, little research has been performed on this molecule [30]. These findings indicate clinical effects of CBD on ASD symptomes.”
At your suggestion, I insered findings of three new review articles on this section maeker by yellow colour on page 14.
10.Modulationof Endocannabinoid Action in Neuroinflammation is related to ASD
Again, describes ASD as neurodegenerative.
My answer: I have changed to neurodevelopmental disorders.
Sections on iron and copper have nothing to do with lipid-based molecules.
My answer: Since ccumulating evidence indicated the the involvement of copper in lipid metabolism, I have inserted following review articles’ findings that “Human copper transporters are subject of a complex multilayer regulation by various metabolic signals (Hasan et al. 2017). ” on pages 20, marker by yellow colours. Therefore, to leave an important role of copper signaling, I have deleted less significnt references, and slso shortened the sentences on this section on page 21.
Conclusion very quickly recaps main points of the article without adding any sort of discussion, synthesis, or new insights. It actually highlights how disconnected a lot of the sections are.
General:
The article focusses on anti-inflammatory pathways (only some of which involve lipid-based molecules), so why does the title choose to emphasize lipid-based molecules? The authors don’t discuss anything about lipid metabolism or why it matters that most of the molecules they’re talking about happen to be lipid-based.
My answer: Lipids participate in the regulation of membrane fluidity, axonal growth, development, memory, and inflammatory response, and significantly effects on various important signing pathways. I have described the important role of lipids and related signaling pathway in the section 2. Signaling Pathways Associated with Lipid Metabolites as follows:
“Lipids act as regulatory molecules that contribute to the growth and maintenance of brain functions. Altered fatty acid metabolic pathways may be involved in ASD pathogenesis. A natural lipid-derived molecule, prostaglandin E2 (PGE2), is involved in the physiological processes involved in ASD [6,7]. Esterified omega-6 polyunsaturated fatty acids (PUFA), such as arachidonic acid (AA), present on the inner surface of the cell membrane, are converted by phospholipase A2 (PLA2) into the free form, which is further metabolized by cyclooxygenases (COXs) and lipoxygenases (LOXs) into bioactive mediators, such as prostanoids and leukotrienes (LTs) [8]. These mediators have been proposed as novel preventive and therapeutic targets for inflammatory diseases, including some types of ASD [8].”on page 5.
Abbreviations are inconsistently used and occasionally redefined.
My answer: I have consistently used corrected and more used abbreviation throughout the text.
Revised figures and legends are not present and cannot be commented
My answer: Editorial staff did not upload the figure 2 and 3. I have uploaded these figures.
The article still lacks focus and still requires re-organization and a clearer message. While it does review the literature, it doesn’t do so in any novel way that hasn’t already been reviewed, and it doesn’t really contribute any clarity to the field.
My answer: This comments are a first severe comments for me. Although this journal has slightly heigh impact factor, I was supprized you such comments
At your adverse criticism, I re-constracred new section on samarizing sentences such as section 12, entitlied title such as “The Association between Lipid Metabolism and Endocannabinoid System and Its Related Signaling Pathways”
If you do not accept my effort, woukd you suggest haw I should make senrences or constraction
I considered that this review provided novel pathophysiology of ASD.
